# Water-Soluble Carbon Quantum Dots Modified by Amino Groups for Polarization Fluorescence Detection of Copper (II) Ion in Aqueous Media

**Anastasia Yakusheva [1,*]** , **Dmitry S. Muratov [1]** , **Dmitry Arkhipov [1]** , **Gopalu Karunakaran [2]** , **Sergei A. Eremin [3]** and **Denis Kuznetsov [1]**

[1] Department of Functional Nanosystems and High-Temperature Materials, National University of Science and Technology MISIS, Leninsky prospect 4, 119049 Moscow, Russia; muratov@misis.ru (D.S.M.); hipa2010@yandex.ru (D.A.); dk@misis.ru (D.K.)

[2] Biosensor Research Institute, Department of Fine Chemistry, Seoul National University of Science and Technology (SeoulTech), Gongneung-ro 232, Nowon-gu, Seoul 01811, Korea; karunakarang5@gmail.com

[3] Faculty of Chemistry, M. V. Lomonosov Moscow State University, Lenin's mountains, 119991 Moscow, Russia; saeremin@gmail.com

* Correspondence: yakusheva.as@misis.ru; Tel.: +7-909-669-30-81

**Abstract:** Industrialization is serious for changing the environment and natural water composition, especially near cities and manufacturing areas. Logically, the new ultrasensitive technology for precise control of the quality and quantity of water sources is needed. Herein, an innovative method of polarization fluorescence analysis (FPA) was developed to measure the concentration of heavy metals in water. The approach was successfully applied for precise tests with reduced analysis time and increased measurement efficiency among laboratory methods. Based on this work, the investigations established the new type of carbon quantum dots (CQDs) with controllable fluorescence properties and functionalized amino—groups, which is appropriate for FPA. The parameters of one and two-step microwave synthesis routes are adjusted wavelength and fluorescence intensity of CQDs. Finally, under optimized conditions, the FPA is showed the detection of copper (2+) cations in water samples below European Union standard (2 mg/L). Moreover, in comparison with fluorescence quenching, polarization fluorescence is proved as a convenient, simple, and rapid test method for effective water safety analysis.

**Keywords:** carbon quantum dots; polarization fluorescent analysis; fluorescence properties; copper cation; water samples

## 1. Introduction

Recently, heavy metals have been shown to be a severe problem due to active industrialization. The problem of environmental pollution is still the main attraction to enter eco-friendly technology. A promising solution is needed for the present tests of toxic chemical elements, compounds, and active ions in the environment sphere. The new alternative technology demands the new investigation and enhancement for keeping up with the changing world [1]. Commonly, many reasons are compared for appropriate research. Measurements will have identified a wide range of chemical compounds, simultaneously with high performance, cost-effective, eco-friendly probe preparation, and elementary experiment stages have not been [SL1] solved completely [2–4].

In the past few decades, the quantitative and qualitative laboratory methods include chromatography, spectrophotometry, solid-phase extraction with atomic absorption spectroscopy, and atomic emission spectroscopy, and mass-spectrometry with inductively coupled plasma for

determining a heavy metal in water probe are generally used [5–10]. However, the methodologies are not so much better for the large-scale laboratory measurements with the growing demand. One of the strategies is to apply functional sensors, which are dedicated to a new measurement method [11]. From this side, quantum dots as optical nanosensors based on zinc, selenium, tellurium, and other chemical element have a greater interest in the studies. Carbon dots are one of the best materials for eco-friendly application. Today, particles are actively investigated as sensors for determining the concentration of chemical compounds in aqueous media. There are so many different types of inorganic nanoparticles with significant drawbacks: expensive precursors for synthesis, insolubility in water, and toxic to the environment [12].

A new promising class of zero-dimensional carbon materials was evaluated as sensors from bio and eco-compatible precursors with fine optical properties [13,14]. Over the past few decades, there has been sustained research activity on carbon quantum dots (CQDs) with a size up to 10 nm [15,16] due to stable and sensitively fluorescent properties [17]. A number of various types of CQDs are still getting bigger over time, but the synthesis stage is common for all [18]. The synthesis of CQDs is carried out in three separate routes: carbonization, passivation, and surface functionalization. The temperature of carbonization is a very sensitive parameter above 300 degrees for the decomposition of carbon-containing precursor [19]. For example, a plant material, such as fruit and nut peels, berries, and juices, have shown the retention of specific functional groups [20] only up to 250 °C on particle surface [21]. Moreover, the fluorescence spectrum and quantum yield of CQDs significantly depend on the carbon and nitrogen ratio in the particle and suffer from inconsistent precursors. Consequently, the preferable synthesis technique for adjusting the first stage is to use chemical compounds instead of natural precursors [22]. Briefly, sulfur, nitrogen, and oxygen as inclusions are mutually involved in nanoparticles for increased quantum yield [23]. Functional groups on the nanoparticle surface are determined by practical application [24,25] in waste-water sensors technology [26]. The type and number of groups influence optical properties [27]. For example, surface adsorption of harmful chemical compounds leads to a strong fluorescence quenching [28,29]. The impurities and functional groups are precisely coordinated contaminant molecules on the surface for excellent sensor properties. While a variety of synthesis routes have been suggested, all its demanding hard-controllable morphology optimization, reduction of surface defects, trap sites, as a result, enhance the fluorescence intensity of passivated carbon nanoparticles.

Usually, the detection of heavy metals in samples is under consideration for concentration cations in aqueous containing probe [30,31]. The measurements based on fluorescent, colorimetric sensors [32,33], and visual detection [34] have many problems with pH tolerance, experiment time, and solvent conditions of actual samples [35].

Herein, polarization fluorescence analysis (FPA) is represented as a new alternative optical measurement. In this perspective, FPA establishes the fluorophore polarization effect, which depends on surface absorption and sample viscosity. The parameters include more proven information about sensitive chemical coordination near the double electron layer of the CQD surface. The measurement data of FPA, in comparison with fluorescence quenching methods, confirm the effectiveness of CQDs nano-sensors to determine copper cations in water.

## 2. Materials and Methods

### 2.1. Reagents and Materials

The precursors used in the synthesis: citric acid (99.8%), ascorbic acid (99.9%), ammonium dihydrogen phosphate (99.0%), and Trilon-B (EDTA-Na$_2$: disodium salt of ethylenediaminetetraacetic acid) (99.5%). The analytical standards of Fe, Co, Ni, Cu, Zn, Ag, Au, Hg, Pb, and Sn, in the range from $1 \times 10^{-5}$ to $1 \times 10^3$ mg/L concentration apply for confirming selective detection of the copper cation in water.

## 2.2. Instruments

Herein, the SUPRA MWS-1803MW microwave oven (Hangzhou, China) was used for the synthesis procedure under 700 W microwave irradiation. The size distribution and morphology of CQDs were characterized by transmission electron microscopy JEOL JEM-1400 (Tokyo, Japan). The size measurements were provided on Malvern Zetasize Nano ZS particle size and particle zeta-potential analyzer ZEN3600 (Malvern Panalytical Ltd, Malvern, UK). The optical properties of CQDs were studied using the Agilent Technologies Cary Eclipse Fluorescence spectrophotometer (Santa Clara, CA, USA) and Thermo Scientific Heλios α spectrophotometer (Waltham, MA, USA). IR-spectra of CQDs were obtained from Thermo Scientific Spectrometer Thermo NICOLET 380 (Waltham, MA, USA) to identify the functional groups. The express fluorescence polarization and intensity tests were conducted at portable fluorimeter Sentry-100 (Waukesha, WI, USA).

## 2.3. Synthesis of CQDs

Firstly, the dry precursor's mixtures were used for the first stage: CQD 1 (0.044 mol of citric acid and 0.022 mol of ammonium dihydrogen phosphate), CQD 2 (0.044 mol of ascorbic acid and 0.022 mol of ammonium dihydrogen phosphate), and CQD 3 (0.044 mol of citric acid, 0.022 mol of ammonium dihydrogen phosphate and 0.004 mol of ethylenediaminetetraacetic acid disodium salt) were heated for 1 min. The fluorescence wavelength and intensity were investigated by adding disodium salt of ethylenediaminetetraacetic acid in the following amounts: 0.004, 0.005, 0.006, 0.007, and 0.008 mol.

For the two-step synthesis route, the 0.044 mol of ascorbic acid and 0.022 mol of ammonium dihydrogen phosphate were irradiated for 1 min, and then 0.004 mol of ethylenediaminetetraacetic acid disodium salt was added to continue reaction in 30 s for CQD 4. Additionally, the variety amount 0.004, 0.005, 0.006, 0.007, and 0.008 mol was added in the second step.

In the experiment, the sample in the crucible was cooled to room temperature and suspended in $3 \times 10^{-2}$ L of water to collect the dark brown dispersion. The next required step is filtering through a Millex-LG 0.2 μm IC filter cartridge (Millipore, Burlington, MA, USA). The $1 \times 10^{-2}$ L of CQDs in a test tube were centrifuged for 30 min on 12.4 g acceleration (14,500 rpm) to remove large particles. More synthesis parameters are shown in Table 1.

**Table 1.** Ratio of precursors and time of microwave irradiation for carbon quantum dots (CQDs) sample.

| S.No. | Sample Code | Citric Acid, mol | Ascorbic Acid, mol | Ammonium Dihydrogen Phosphate, mol | 1st Step Microvalve Irradiation, s | Trilon-B (EDTA-Na$_2$), mol | 2nd Step Microvalve Irradiation, s |
|---|---|---|---|---|---|---|---|
| 1. | CQD 1 | 0.044 | - | 0.022 | 60 | - | - |
| 2. | CQD 2 | - | 0.044 | 0.022 | 60 | - | - |
| 3. | CQD 3_1 | 0.044 | - | 0.022 | 60 | 0.004 | - |
| 4. | CQD 3_2 | 0.044 | - | 0.022 | 60 | 0.005 | - |
| 5. | CQD 3_3 | 0.044 | - | 0.022 | 60 | 0.006 | - |
| 6. | CQD 3_4 | 0.044 | - | 0.022 | 60 | 0.007 | - |
| 7. | CQD 3_5 | 0.044 | - | 0.022 | 60 | 0.008 | - |
| 8. | CQD 4_1 | 0.044 | - | 0.022 | 60 | 0.004 | 30 |
| 9. | CQD 4_2 | 0.044 | - | 0.022 | 60 | 0.005 | 30 |
| 10. | CQD 4_3 | 0.044 | - | 0.022 | 60 | 0.006 | 30 |
| 11. | CQD 4_4 | 0.044 | - | 0.022 | 60 | 0.007 | 30 |
| 12. | CQD 4_5 | 0.044 | - | 0.022 | 60 | 0.008 | 30 |

## 2.4. Detection of Metal Ions

The FPA is establishing a technique of measurements that considers surface absorption and sample viscosity more informative in comparison with fluorescence intensity. According to the results, CQD with constant fluorescence emission was preferable for measurement on portable fluorimeter

SENTRY-100. The optical system of fluorimeter is poorer than laboratory equipment due to the optical detector on 535 nm wavelength emission.

In experiments, the optical parameters of CQDs calculate for concentration from 0.1 mg/L to 20 mg/L in dispersion. In addition, the fluorescence quenching confirms on Agilent Technologies Cary Eclipse Fluorescence spectrophotometer. The results fluorescence intensity (FI) of CQD with different positions of the maximum emission spectra were chosen as the most appropriate particles and compared with fluorescence intensity data on SENTRY-100 in Figure 1.

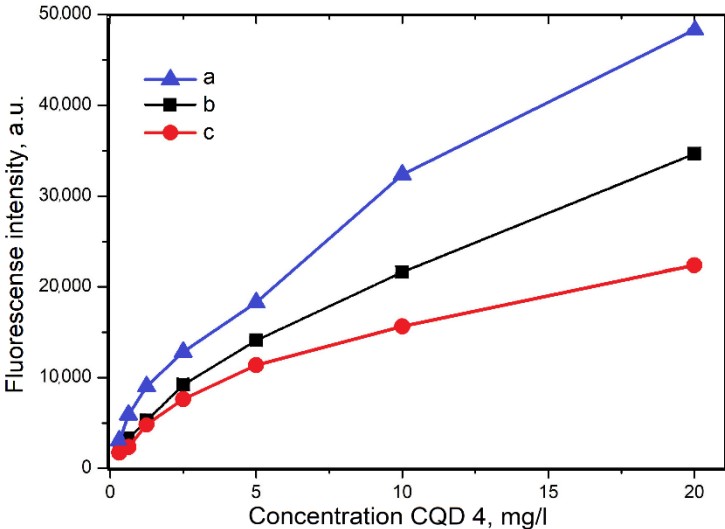

**Figure 1.** Fluorescence intensity of CQD 4 depends on the concentration in deionized water. The emission peak is 530 nm (**a**), 560 nm (**b**), and 500 nm (**c**), respectively.

As reported before, the fluorescence intensity of CQD 4 was better for samples with the position of fluorescence peak at 530 nm due to properties of CQD (530 nm) are more closest to the SENTRY-100 optical system, then 500 nm (Figure 1c) and 560 nm (Figure 1b). Moreover, the higher fluorescence intensity may apply to reduce the concentration of CQDs in the measurement cuvette and also increase the experiment sensitivity.

In this case, the CQD dispersion (the position of maximum fluorescence at 530 nm) with a final concentration at 6.8 mg/L was designed, which corresponds to recommended fluorescence intensity 24,000 a.u. to achieve the maximum accuracy for FI and FP. The technique of metal cations concentration measuring was studied in standard cations solutions of Fe, Co, Ni, Cu, Zn, Ag, Au, Hg, Pb, and Sn. The concentration of metal cations was used from $10^{-5}$ mg/L to $10^3$ mg/L and added in a volume of $2 \times 10^{-5}$ L in mL CQD's sample.

## 3. Results and Discussion

### 3.1. Characterization of CQD

In the present study, the better alternative microwave assistance synthesis of amino-functionalized CQDs present. The order to evaluate the optical properties depend on the synthesis route [36], the influence of precursors (a type of carbon source) was identified in Figure 2.

The core of fluorescence CQD 1 and CQD 2 with a varied source for carbonization stage correspond with acid sources and position of absorption (ABS) peak. Briefly, in the equal amount of citric acid or ascorbic acid, the ABS peak of CQD 1 at 250 nm, and the peak of CQD 2 is attributed to the right on 90 nm. The evolution between the Stokes shift of the first and second type of nanoparticles is slightly different. The fluorescence (FL) peak excited at a maximum of ABS wavelength shown in Figure 2B at 410 nm and 500 nm for CQD 1 and CQD 2, respectively. Optical curves have been in equal intensity of excitation and emission.

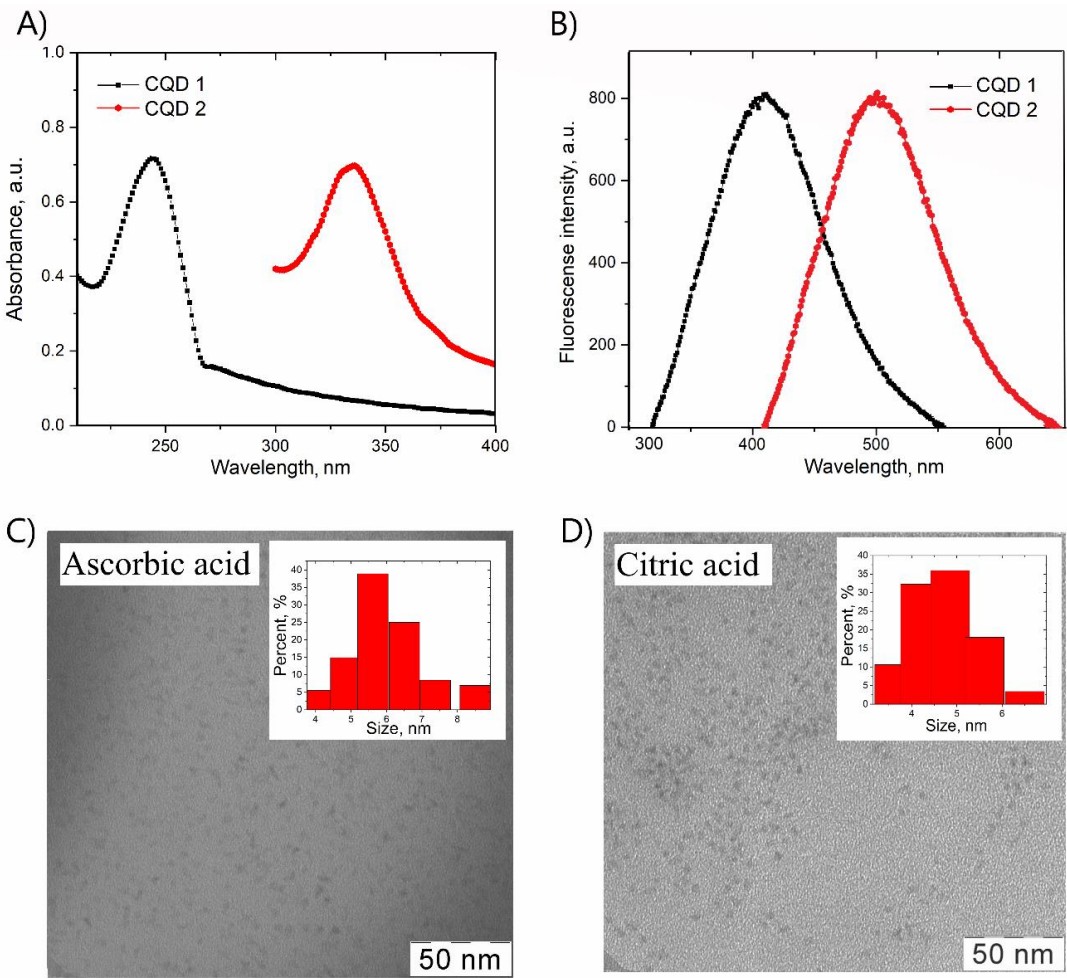

**Figure 2.** Characterizations carbon core of CQD 1, CQD 2, (**A**) ABS curve CQDs from ascorbic or citric acid as carbon source, (**B**) FL spectra of CQD 1 and CQD 2, corresponding with ABS spectra, (**C**) TEM image and size distribution of CQD 1, and (**D**) TEM image and size distribution of CQD 2.

The morphology of nanoparticles and size distribution assessment provide from the TEM image. Pictures showed similar quasi-spherical CQDs with size near 5 nm at both synthesis routes. Noticeably, the average size for each type of dot is not correlating to FL peak position due to relative Stock shift at ABS spectra attribute to the structure of the carbon bond in precursors [37].

### 3.2. Fluorescence Properties of CQD

In this case, the carbon source determined the position of maximum fluorescence as a limit factor for CQDs design. The acids for CQD were saved in a molar ratio to ammonium dihydrogen phosphate (oxidizer with phosphorus and nitrogen inclusions). The constant ratio supports the uniform fluorescent core of carbon dots in the synthesis condition. Secondly, a similar effect was confirmed at the amount of EDTA-Na$_2$, if comparing CQD 3 and CQD 4, which was shown in spectra in Figure 3A–D, respectively. In comparison, the ability of two-steps microwave irradiation to improve fluorescence properties proven here.

As exhibited in spectra CQD 3 and CQD 4 in Figure 3A,C, the two-steps microwave irradiation plays a key role in surface energy formation. In particular, the excitation in Figure 3C has attracted attention to absorbance shift.

Moreover, in Figure 3B,D, the FI of CQD 4 is intensive twice more than CQD 3 and shows the green and yellow colored light. Briefly, the CQD 3 represented a maximum emission wavelength of 535 nm under 380 nm of exciting wavelength and all emission spectra in the 500–535 nm range. Opposite the

CQD 4 shows the broad range of emission from 510 to 570 nm and a maximum emission wavelength of 570 nm under 400 nm of exciting wavelength. The key factor of the emission dependence is associated with the coordinating center in EDTA-Na$_2$ [38], which is safe for the structure and evenly distributed on the particle surface. Consequently, the two-step microwave synthesis CQD 4 was shown to have a better ability for fluorescent intensity, which can be controlled by changing the type and number of precursors for functionalization.

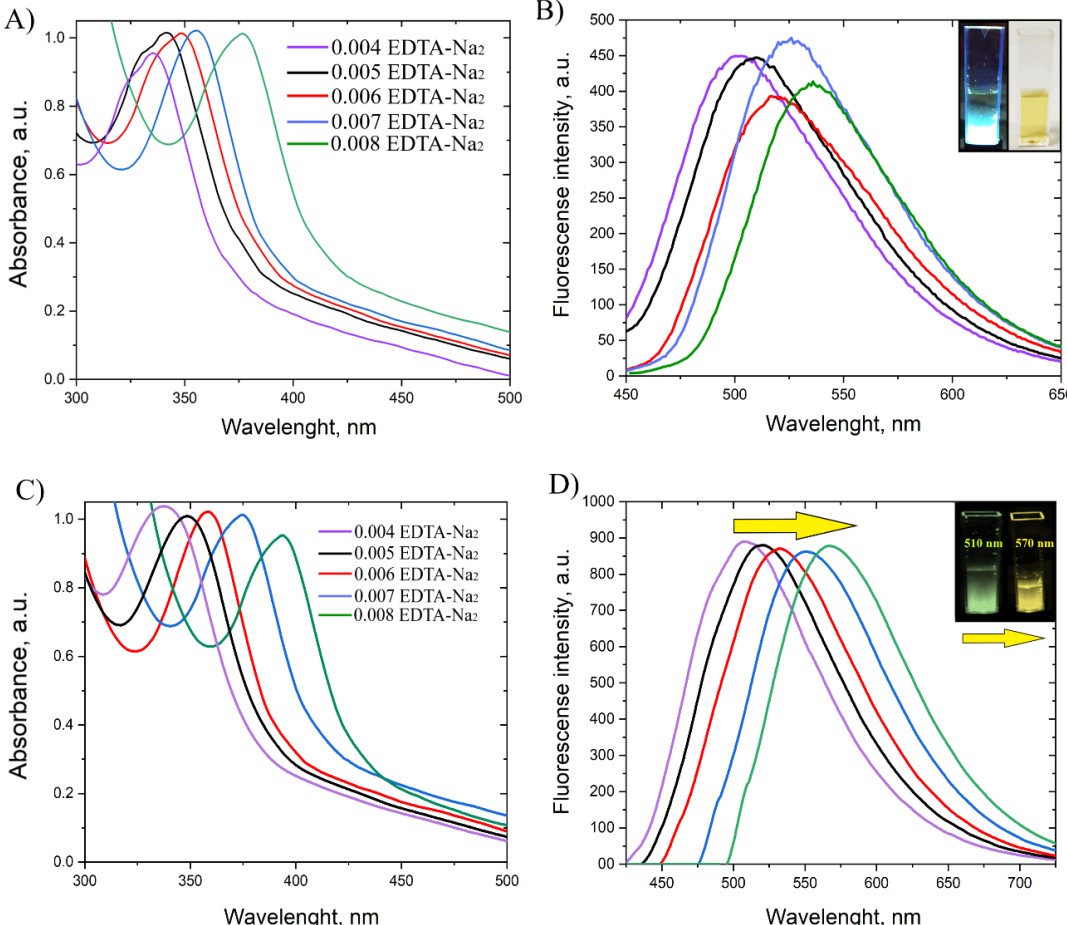

**Figure 3.** (**A**) Absorbance spectra of CQD 3 with a different mol EDTA-Na$_2$ (0.004, 0.005, 0.00. In this case, the CQD dispersion (the position of maximum fluorescence at 530 nm) with a final concentration at 0.68 mg/L was designed, 0.007, 0.008), (**B**) emission spectra of CQD 3, respectively, (**C**) absorbance spectra of CQD 4 with a different mol EDTA-Na$_2$ (0.004, 0.005, 0.006, 0.007, 0.008), (**D**) emission spectra of CQD 4.

Size Distribution of CQD 1, 3 and 4

To explore fluorescence behavior, secondly, the size distribution of CQD was measured by a dynamic light scattering method in Figure 4.

The size distribution exhibited the structure of the nanoparticles and affected to fluorescence properties of quantum dots [39]. Herein, the curves of size distribution in Figure 3A correspond with three types of CQD 1, CQD 3.3, and CQD 4.3 particles, respectively. These are attributed to the core (CQD 1), the average size was near 3–4 nm. If EDTA-Na$_2$ is added during the synthesis rote, the average size increases to 6–7 nm for the one-step process and 8–9 nm for the two-step process, as shown in Figure 3B, due to consecutive carbonization of precursors. The results show that the average size of CQDs was effectively changed by steps of synthesis (in particular, the time of microwave irradiation) [40].

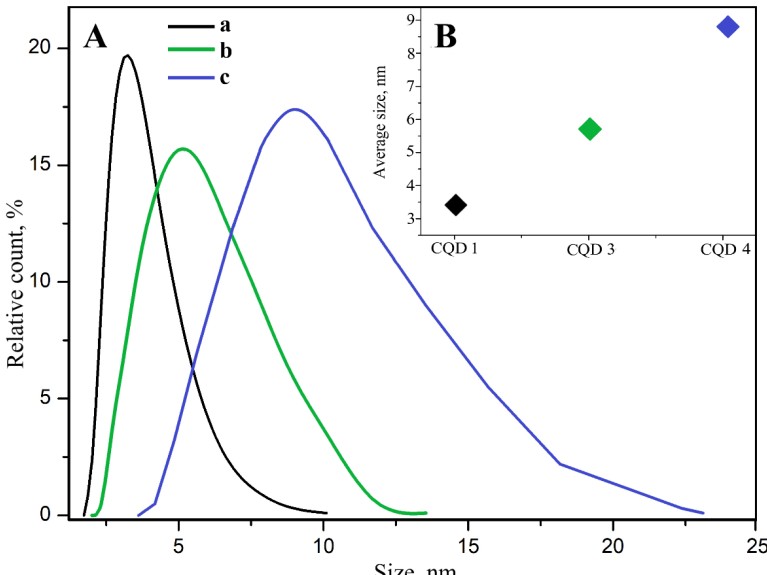

**Figure 4.** (**A**) Size distribution of CQD 1, CQD 3, and CQD 4. (**B**) The average size of CQD 1, CQD 3 and CQD 4.

### 3.3. FI Quench of CQD by Addition of Metal Ions

The technique of metal cations concentration measuring by CQD 4 with appropriate controllable optical properties was studied in standard cations solutions of Fe, Co, Ni, Cu, Zn, Ag, Au, Hg, Pb, and Sn. The results of fluorescence quenching for metal salt solutions were presented in Figure 5.

The bar charts in Figure 5A show that $5 \times 10^{-5}$ L all-metal salt solution with $10^{-3}$ L CQD 4 dispersion close to fluorescence quenching to pure CQDs solution. In opposite to previous metals, fluorescence intensity dramatically decreased after $Cu^{2+}$ cations-coordinating reaction at CQD 4. In particular, the result established reaction activity between nanoparticle and cupper cations [41]. As shown in Figure 5B,D, cupper cations can effectively quench the fluorescence of CQD 4 from $1 \times 10^{-4}$ mg/mL to 1 g/mL. During the experiment, the accuracy of measurements is confirming by the added-found approach. Fluorescence quenching demonstrated an interaction between the copper cation and CQD.

First of all, the exploration of chemistry starts with FTIR analysis. The method was used as an indicator to optimize measurements of $Cu^{2+}$ in water samples. The FTIR spectra in Figure 5C. are shown the functional groups on CQD 4 before adding the $Cu^{2+}$ solution Figure 5Ca and after Figure 5Cb. The CQD 4 reactions on the surface are characterizing by the peaks at 3420 $cm^{-1}$ and 1420 $cm^{-1}$ due to the change in line intensity of hydroxyl groups. A peak at 1700 $cm^{-1}$ is attributed to the carboxyl group, which is retained from citric acid. The peak at 1650 $cm^{-1}$ was correlated with the valence $C = N$ bond and amino-groups on the surface of nanoparticles correspond with the peak of 1095 $cm^{-1}$. Carbon- carbon bonds, like $C − C$ and $C = C$, are reflected by peaks of 1360 $cm^{-11}$, 1230 $cm^{-1}$. In summary, the IR-spectra, the hydroxyl, carboxyl, and amino groups on the CQDs surface were found. Copper was causing an intensity change in the spectrum, which depend on the chemical bonds in functional groups. The greatest change corresponds with $Cu^{2+}$ and amino group reaction via non-fluorescence complexes formation [42,43].

The concentration of copper ions was investigated in both: the FPA and FI quenching method, in comparison by Sentry-100 measurements. The data of fluorescence polarization correlated with the average angular displacement of the fluorophore, which occurs absorption to emission photon. This value depends on the rotational diffusion of a fluorophore, which is determined by the viscosity of the sample, temperature, and particle size. When viscosity and temperature are constant, the fluorescence polarization shows particle size changes due to copper is joined to the CQD surface. The ratio FI, FP, and concentration of copper cations are presented in Figure 6.

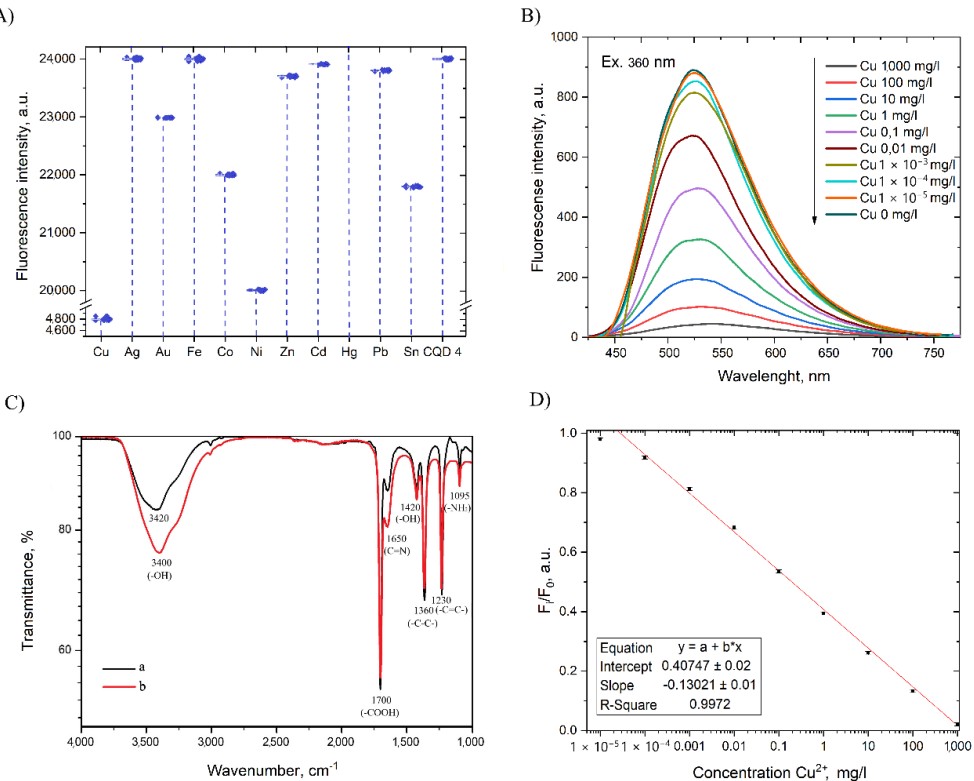

**Figure 5.** (**A**) The fluorescence quenching effect of CQD 4 with metal cation in water, (**B**) Concentration dependence for the fluorescence quenching of PL CQD 4 by $Cu^{2+}$, (**C**) FTIR spectra of CQD 4 before (a) and after (b) coordinated $Cu^{2+}$ cations, and (**D**) The linear relationship between the fluorescence intensity CQD 4 and the $Cu^{2+}$ concentration.

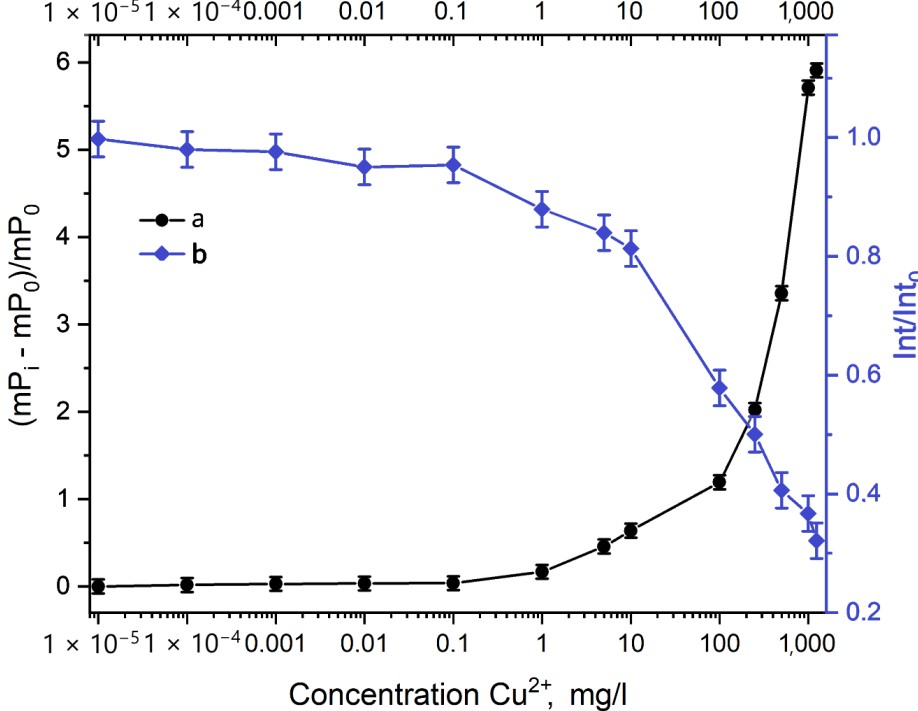

**Figure 6.** Relative changes between fluorescence polarisation (**a**) and fluorescence intensity (**b**) of CQD 4 sample with different copper (2+) cations concentration.

For FPA rapid analysis (Figure 6a) low detection limit of $Cu^{2+}$ concentration is 1 mg/L, which is comparable to fluorescence intensity (Figure 6b) measurements from outside laboratory location [44]. The method of measuring the fluorescence polarization allows determining concentrations, which are compared with the fluorescence intensity method, but the errors show a high accuracy level. In addition, functional groups react with copper ions, hence, the average size of carbon nanoparticles was increased. The relative change in polarization values shows a fivefold data increase, opposite to fluorescence intensity is limited to a range of measured concentrations. At the same, the relative measurement error is reduced when using the method of FPA, which is more suitable for the determination of copper.

In particular, a calibration graph for FPA of $Cu^{2+}$ concentration and fluorescence polarization of CQD 4 is presented in Figure 7.

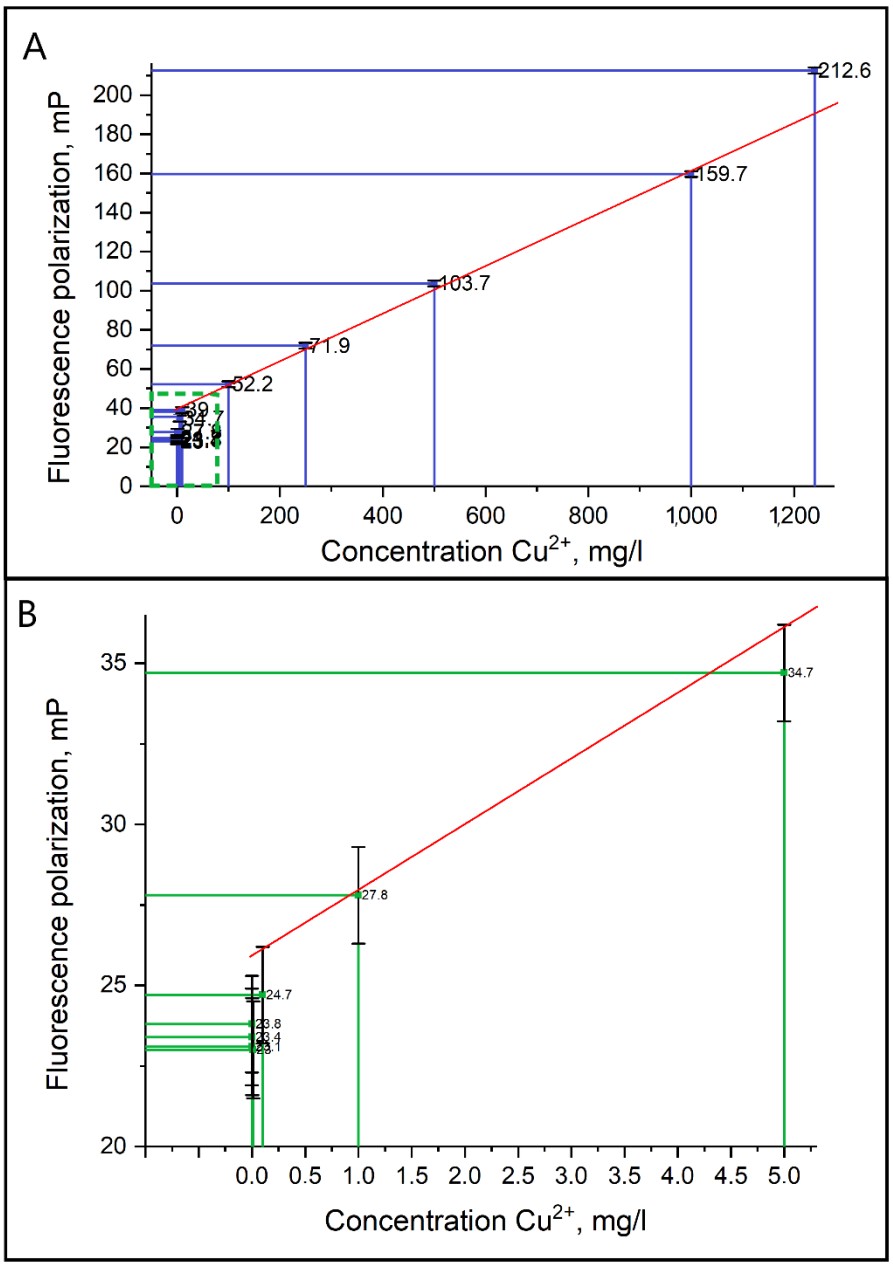

**Figure 7.** Calibration graph (A) CQD 4 fluorescence polarization values and copper (2+) cation concentrations in the water sample from 1 mg/L to 1240 mg/L, (B) Up-scale graph for 0 mg/ml to 5 mg/ml copper (2+) cation concentrations

A detection of limit in a water sample (7.7 μmol/L) investigated via the added-found method when the linear relationship between the quantities from 1 mg/L to 1g/L ($2 \times 10^{-5}$ L added in a sample) is constant. For comparative, standards of safe copper concentration in natural water 2 mg/L (WHO, EU). Consequently, the FPA can be used to determine the safety and ecological state of water resources.

## 4. Conclusions

Recent studies have shown efficient two-step microwave-assisted synthesis of CQD. Nanoparticles with improved optical properties and fluorescence maximum from 510 to 570 nm get via changing the precursor amount at the second step of the approach. The amino groups on the surface of CQD 4 allowed the use of carbon nanoparticles for selective research of copper cations in aqueous samples from 1 mg/L to 1 g/L. Confirming result is based on polarization fluorescence analysis, which admitted to the measurement of a hundred simultaneous water intake samples. One of the advantages of the method is the measurement ability for hard-to-reach or remote water sources. In conclusion, the time to time screening of water safety and assessment of copper cations at the appropriate level, which satisfactorily meets the limit of $Cu^{2+}$ ions in drinking water (2 mg/L EU standard), was proven by FPA based on CQD 4.

**Author Contributions:** Conceptualization, A.Y., D.K., S.A.E.; Methodology, A.Y., S.A.E.; Software, D.S.M.; Validation, G.K., S.A.E., D.S.M.; Formal Analysis, A.Y.; Investigation, A.Y.; Resources, D.K., S.A.E., D.A.; Data Curation, A.Y.; Writing—Original Draft Preparation, A.Y., D.S.M.; Writing—Review and Editing, G.K.; Visualization, D.A.; Supervision, D.K.; Project Administration, D.K. Funding Acquisition, D.A. All authors have read and agreed to the published version of the manuscript.

**Funding:** The authors gratefully acknowledge the financial support of the Russian Foundation for Basic Research according to the research project No. 18-29-25051.

**Conflicts of Interest:** The authors declare no conflict of interest.

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
