# Peer review of "Water-Soluble Carbon Quantum Dots Modified by Amino Groups for Polarization Fluorescence Detection of Copper (II) Ion in Aqueous Media"

_processes, doi:10.3390/pr8121573_

Round 1

Reviewer 1 Report

By using two steps microwave irradiation technique and changing the precursor amount at the second step,the carbon quantum dots(CQDs) with improved optical properties were synthesized for detection of copper (2+) cations in water samples based on polarization fluorescence analysis. The manuscript showed the rigorous design of experiment, but has some drawbacks that should be addressed:

  1. In part 2.4, why did you choose 0.68 mg/L as final concentration and how did you define the maximum accuracy? In addition, the corresponding fluorescent intensity of 0.68 mg/L CQD in Figure 1 is not 24000 a.u.. Please check it out.
  2. Compared with fluorescence intensity of CQD 1 (about 800 a.u.) in Figure 2B which did not be treated with EDTA-Na2, why did the fluorescence intensity of CQD 3 after adding the EDTA-Na2 decrease dramatically to 450 a.u. in Figure 3B?
  3. In Figure 4, the article characterized that average size of CQD 3 and CQD 4. Was it the mixture of all CQD 3/CQD 4 with different amount of EDTA-Na2 or a specific one for example CQD 4(0.004 EDTA-Na2)? Please provide the details for easier understanding.
  4. Some minor issues should be checked carefully, such as, in Part 3.2, “As described in Fig 2B and C” should be corrected to “Fig 3B and D”, and “EDTA-NA2” should be uniform to “EDTA-Na2”.

Author Response

Dear reviewer,

Thank you for comments and suggestions.

The more detailed answer in file.

Reviewer 2 Report

This manuscript “Water-Soluble Carbon Quantum Dots Modified by Amino Groups for Polarization Fluorescence Detection of Copper (II) Ion in Aqueous Media” authored by Anastasia Yakusheva et al. reported the development of polarization fluorescence analysis for the detection of copper (II) ion in aqueous media using carbon quantum dots. This technique that they developed showed excellent sensitivity and broad linear range for copper (II) ions with detection limit below 2mg/ml. This method is simple and fast, which is suitable for water safety analysis. I would recommend this manuscript to be published in the journal “Processes” with some minor points need to be corrected as listed below.

1. Typo: On page 6, “As described in Fig 2B and C the FI of CQD 4 is intensive in twice then CQD3”. The figure number should be Fig “3B” and “D”.

2. What’s the main difference between using citric acid or ascorbic acid in the synthesis of CQD?

3. How is the stability of this polarization fluorescence detection technique? Is it pH sensitive?

Author Response

Dear reviewer,

We would like to thank you for having our manuscript revise.

The necessary corrections have been carried out in this dounload manuscript.

We attach our detailed answers in the file.
